# The Poly-Arginine Peptide R18D Interferes with the Internalisation of α-Synuclein Pre-Formed Fibrils in STC-1 Enteroendocrine Cells

**DOI:** 10.3390/biomedicines11082089

**Published:** 2023-07-25

**Authors:** Anastazja M. Gorecki, Holly Spencer, Bruno P. Meloni, Ryan S. Anderton

**Affiliations:** 1School of Health Sciences, University of Notre Dame Australia, Fremantle, WA 6160, Australia; holly.spencer@health.wa.gov.au (H.S.);; 2School of Biological Sciences, University of Western Australia, Crawley, WA 6009, Australia; 3Perron Institute for Neurological and Translational Science, Nedlands, WA 6009, Australia; 4Centre for Neuromuscular and Neurological Disorders, The University of Western Australia, Nedlands, WA 6009, Australia; 5Department of Neurosurgery, Sir Charles Gairdner Hospital, First Floor, G-Block, QEII Medical Centre, Nedlands, WA 6008, Australia

**Keywords:** α-synuclein, arginine, poly-arginine, R18D, enteroendocrine cells, pre-formed fibrils

## Abstract

In Parkinson’s disease (PD), gut inflammation is hypothesised to contribute to α-synuclein aggregation, but gastrointestinal α-synuclein expression is poorly characterised. Cationic arginine-rich peptides (CARPs) are an emerging therapeutic option that exerts various neuroprotective effects and may target the transmission of protein aggregates. This study aimed to investigate endogenous α-synuclein expression in enteroendocrine STC-1 cells and the potential of the CARP, R18D (18-mer of D-arginine), to prevent internalisation of pre-formed α-synuclein fibrils (PFFs) in enteroendocrine cells in vitro. Through confocal microscopy, the immunoreactivity of full-length α-synuclein and the serine-129 phosphorylated form (pS129) was investigated in STC-1 (mouse enteroendocrine) cells. Thereafter, STC-1 cells were exposed to PFFs tagged with Alexa-Fluor 488 (PFF-488) for 2 and 24 h and R18D-FITC for 10 min. After confirming the uptake of both PFFs and R18D-FITC through fluorescent microscopy, STC-1 cells were pre-treated with R18D (5 or 10 μM) for 10 min prior to 2 h of PFF-488 exposure. Immunoreactivity for endogenous α-synuclein and pS129 was evident in STC-1 cells, with prominent pS129 staining along cytoplasmic processes and in perinuclear areas. STC-1 cells internalised PFFs, confirmed through co-localisation of PFF-488 and human-specific α-synuclein immunoreactivity. R18D-FITC entered STC-1 cells within 10 min and pre-treatment of STC-1 cells with R18D interfered with PFF uptake. The endogenous presence of α-synuclein in enteroendocrine cells, coupled with their rapid uptake of PFFs, demonstrates a potential for pathogenic spread of α-synuclein aggregates in the gut. R18D is a novel therapeutic approach to reduce the intercellular transmission of α-synuclein pathology.

## 1. Introduction

Parkinson’s disease (PD) is a chronic neurodegenerative disorder that presents with vast clinical and neuropathological disease heterogeneity. Growing evidence demonstrates distinct disease trajectories, as some individuals predominantly present with motor symptoms, while others report gastrointestinal symptoms (e.g., constipation, nausea, bloating) or other non-motor symptoms [1,2,3], sometimes years prior to motor symptom onset and clinical diagnosis [4,5]. Increased understanding of the multi-faceted and bidirectional communication between the gut and brain has led to the notion of ‘top-down’ and ‘bottom-up’ PD progression via the gut–brain axis [6,7]. There are still no effective therapies for PD and those diagnosed with the disorder face an inevitable deterioration in motor abilities, cognitive function and overall health and quality of life.

Despite the clinical heterogeneity, intraneuronal protein aggregates (termed Lewy bodies) are a defining and unifying feature of PD. The main component of pathogenic Lewy bodies is α-synuclein, a natively unfolded protein with immense conformational flexibility [8]. It exists as monomers and oligomers in healthy cells, where it is involved in neurotransmitter release, synaptic transmission and mitochondrial and lysosomal function [9]. The misfolding of α-synuclein occurs due to a complex and dynamic interaction between environmental, genetic and age-related factors, which can alter proteostasis mechanisms. Over time, these alterations facilitate the formation of insoluble α-synuclein aggregates, which are linked to various downstream cascades resulting in mitochondrial damage, synaptic dysfunction and cell death [10]. The gut and gut microbiome are significantly altered throughout the lifespan by environmental and lifestyle factors [11,12] and numerous studies demonstrate that some people with PD exhibit altered composition and function of the gut microbiome [13,14,15,16] and a leaky and inflamed gut [17,18,19,20]. Importantly, such gut dysfunction is hypothesised to trigger α-synuclein aggregation in the gut [21], particularly in people with slow-progressing PD, bilateral symptom presentation, gastrointestinal symptoms and cognitive decline [6,22,23]. Supporting the notion of bottom-up PD pathology, various rodent models demonstrate that aggregated α-synuclein can spread from the gut to the central nervous system (CNS) via the gut-brain axis in a ‘prion-like’ fashion [24,25,26,27], while gut dysfunction can cause or exacerbate motor symptoms [28,29]. However, these studies relied on exogenous administration of pre-formed α-synuclein fibrils (PFFs) and endogenous sources of α-synuclein pathology remain understudied. In the gut, α-synuclein is expressed by enteric neurons [30] and a handful of studies also report expression by enteroendocrine cells in vivo and in vitro [31,32,33]. In addition, the gut epithelium is an important site of host-microbial interaction, where pro-inflammatory factors may alter the physio-chemical environment and contribute to α-synuclein protein aggregation in PD. 

Given the complex and heterogenous nature of PD, the search for effective disease-modifying therapies has been elusive. Targeting the gut may provide a window to interrupt early pathogenic processes and slow disease progression, especially considering the gut is more readily manipulated by oral medications than the CNS. One interesting and under-studied area of therapeutic potential is the application of cationic arginine-rich peptides (CARPs) for PD. These positively charged peptides can traverse cell membranes to enter cells and can exert neuroprotective effects by maintaining cytosolic calcium homeostasis, preventing excitotoxicity and mitochondrial damage, as well as having anti-microbial and anti-inflammatory properties (reviewed recently [34]). In addition to their established neuroprotective actions, CARPs can inhibit protein aggregation (reviewed recently [35]). For example, different poly-arginine and arginine-rich peptides inhibit tau protein and amyloid-beta peptide aggregation. Therefore, the poly-arginine peptide R18D (18-mer of D-arginine; net charge +18) with its confirmed neuroprotective actions [36] may have the capacity to prevent α-synuclein protein aggregation and/or cellular uptake of aggregates while providing neuroprotection from existing PD pathology. As such, this study aimed to investigate endogenous α-synuclein in enteroendocrine cells, a gut barrier cell type that interacts with the microbiome and the potential of R18D to prevent the internalisation of pathogenic α-synuclein aggregates in enteroendocrine cells.

## 2. Materials and Methods

### 2.1. Cell Culture

#### Enteroendocrine Cell Cultures

The intestinal secretin tumour (STC-1) cell line (ATCCRL3254; In vitro *technologies*), derived from a mouse small intestinal neuroendocrine carcinoma, is an accepted model for enteroendocrine cells [37,38]. Cells were maintained in 25 cm^2^ culture flasks in a CO_2_ incubator (5% CO_2_, 95% air balance, 98% humidity, 37 °C) in Dulbecco’s modified Eagle medium (DMEM; Invitrogen) with 15% horse serum (#26050088, Thermo Fisher Scientific, Australia), 2.5% foetal bovine serum (FBS; #35-076-CV, Corning) and 30 mg/mL each of streptomycin and penicillin, as previously established [39]. Cells were passaged every 2–3 days (at 90% confluency) by trypsinisation (TrypLE Express; Thermo Fisher Scientific) and studied between passages 25–40 [38]. For immunocytochemistry, STC-1 cells were seeded at a density of 150,000 cells per well onto poly-D-lysine (70–150 kDa; Sigma Aldrich, Australia) coated glass coverslips (13 mm diameter) in a 24-well plate and used 1–3 days after plating. 

### 2.2. Cell Stimulation

#### 2.2.1. Pre-Formed α-Synuclein Fibrils

To investigate the effect of aggregated α-synuclein on STC-1 cells in vitro, PFFs were generated from recombinant human α-synuclein monomer protein (RP-003, Proteos, Kalamazoo USA) as previously validated [40]. The α-synuclein protein was diluted to 1 mg/mL in PBS prior to incubation at 37 °C with shaking at 150 rpm for 7 days and subsequently sonicated in 10 s bursts. PFF formation was confirmed using Thioflavin-T assay [40], compared against α-synuclein monomer (RP-001, Proteos) and PBS, with fluorescence measured at 450 nm excitation and 485 nm emission on a Cytation 5 Cell Imaging Multimode Reader (BioTek, #M-2129). Mean fluorescence was compared using one-way analysis of variance (ANOVA) and Tukey’s multiple comparisons test, with *p* < 0.05 considered significant, in GraphPad Prism version 9.4.1 for Mac OS X (GraphPad Software, Boston, MA, USA).

A subset of PFFs were labelled (PFF-488) using the Alexa Fluor 488 Microscale Protein Labelling Kit as per the manufacturer’s instructions (A30006; Thermo Fisher Scientific). For experimentation, PFF-488 (0.2 μg/μL in media) was added to cells and incubated for 2 or 24 h in the CO_2_ incubator, then fixed for immunocytochemistry. To track the presence of PFF-488, live cells were imaged using a fluorescent microscope at 20× magnification at the experiment endpoint prior to fixation.

#### 2.2.2. R18D Treatment of STC-1 Cells

R18D (H-rrrrrrrrrrrrrrrrrr-OH; r = D-arginine) and R18D with fluorescein isothiocyanate (FITC) tag (R18D-FITC; FITC-rrrrrrrrrrrrrrrrrr-OH) were synthesised by Mimotopes (Mulgrave, Victoria, Australia). Peptides were purified via high-performance liquid chromatography to attain 95% purity and prepared as 500 μM stocks in Baxter water. STC-1 cell cultures were treated with R18D or R18D-FITC (5 or 10 μM) diluted in culture media for 10 min at 37 °C prior to the addition of PFF. The R18D or R18D-FITC peptides remained in the culture medium during a 2 h exposure of STC-1 cells to PFFs, but peptide concentration was reduced by 50% after the addition of PFF. 

##### MTS Assay

For viability assessments, cells were plated in a 96-well plate at 45,000 cells per well in 100 µL and incubated with various concentrations of R18D (0.5, 1, 2, 5, 10 µM; 6 wells per group) for 24-h. At the endpoint, 20 µL of 3-(4,5, dimethyliazol-2-yl)-5-(3 carboxymethoxyphenyl)-2-(4-sulfophenyl)-2H-tetrazolium salt (MTS; Promega, Australia) was added to each well. After incubation in a CO_2_ incubator for 90 min, absorbance was measured at 490 nm using a plate spectrophotometer. MTS absorbance data were converted to reflect proportional cell viability relative to both the untreated control (no peptide) (assumed 100% viability). Mean fluorescence was compared in GraphPad Prism version 9.4.1 using one-way ANOVA and Dunnett’s multiple comparisons tests (comparing to the control mean), with *p* < 0.05 considered significant. 

### 2.3. Microscopy

#### 2.3.1. Immunocytochemistry

For immunocytochemistry, glass coverslips (13 mm diameter) were placed in a 24-well plate and coated with poly-D-lysine (500 μL/well) for 1 h prior to cell seeding. At the experiment end point, cells were fixed with 4% paraformaldehyde (PFA) for 10 min at room temperature. Coverslips were then washed with PBST (PBS with 0.1% Tween 20) before blocking in 10% goat serum (#16-210-064, Thermo Fisher Scientific) and 0.1% bovine serum albumin in PBST at room temperature for 1 h. Coverslips were incubated overnight at 4 °C in blocking solution with the following primary antibodies: α-synuclein (#212184; Abcam; Melbourne, VIC Australia), α-synuclein phosphorylated at serine 129 (pS129, #825701; BioLegend; purchased from Australian Biosearch, Wangara WA Australia), human-specific α-synuclein (#MJFR1; Thermo Fisher Scientific), α-tubulin (#15246; Abcam) or β-tubulin (#MA5-16308; Thermo Fisher Scientific). The following day, coverslips were washed in PBS three times for 5 min and incubated at room temperature for 1 h with appropriate secondary antibody in blocking solution: Alexa Fluor 488 goat anti-rabbit (#A32731; Thermo Fisher Scientific), Alexa Fluor 488 goat anti-mouse (#ab150117; Abcam), Alexa Fluor 568 goat anti-rabbit (#A11011; Invitrogen), or Alexa Fluor 568 goat anti-mouse (#ab175701; Abcam), prior to incubation with DNA stain DAPI (1:1500; #10236276001; Sigma) in PBS. Coverslips were mounted onto glass slides using Prolong Diamond Antifade Mountant (#P36970; Invitrogen).

#### 2.3.2. Confocal Microscopy

Slides were imaged using a Nikon Ti-E inverted motorised microscope with Nikon A1Si spectral detector confocal system using a Plan Apo VC 100× NA1.4 oil immersion objective lens (Nikon Corporation, Sydney, Australia) with a 40.0 μm pinhole radius. Slides were excited with 405 nm (violet), 488 nm (blue) and 568 nm (yellow) lasers. Z-stacks were acquired by setting top and lower limits of the stack based on cellular processes as visualised by α-tubulin or β-tubulin, with image slices taken every 0.5 μm at 100× magnification. Images were acquired using NIS-Elements AR (Nikon). For z-stacks, at least two random fields of view on the x–y plane were captured per sample, with images collected from two independent experimental replicates. Representative z-stack images are displayed as a composite image of maximum projection intensity or the orthogonal view. For semi-quantification of PFF uptake, the fluorescence intensity of human α-synuclein staining was calculated from a single image plane and normalised to the cell count per image (based on nuclei number as per DAPI staining), using Fiji [41]. At least four images were quantified, with an average of 56 cells per image. The normalised fluorescence intensity was compared between groups using the Kruskal–Wallis and Dunn’s multiple comparisons test, with *p* < 0.05 considered significant, in GraphPad Prism version 9.4.1. 

## 3. Results

### 3.1. STC-1 Cells Express α-Synuclein

Immunohistochemical staining for α-synuclein and pS129 in STC-1 cells indicated distinct staining patterns (Figure 1). In STC-1 cells, nuclear staining was observed for α-synuclein (maximum intensity projection demonstrated in Figure 1A and orthogonal views in Figure 1C). Conversely, STC-1 cells displayed strong immunoreactivity for pS129 α-synuclein in fibrillar-like formations along cytoplasmic processes and in perinuclear areas (Figure 1B,D). Given the association of phosphorylated α-synuclein with protein aggregation and the reported proximity of enteroendocrine cells to enteric neurons in the gut, we next investigated α-synuclein PFFs in STC-1 cells.

### 3.2. PFFs Enter STC-1 Cells

PFF formation was confirmed through a thioflavin-T assay. Relative to the PBS control, PFFs demonstrated significantly greater fluorescence (94.78 ± 1.25) than both PBS controls (1.00 ± 0.00, *p* < 0.0001) and untagged monomeric α-synuclein (1.09 ± 0.01, *p* < 0.0001) (Figure 2A). Successful tagging of α-synuclein with GPF was confirmed by staining for human α-synuclein after two hours of PFF-488 incubation with STC-1 cells, which demonstrated the majority of PFF-488 immunoreactivity co-localised with human α-synuclein (Figure 2B). Moreover, immunostaining with human α-synuclein did not show a positive signal to the endogenous protein in STC-1 cells (Figure 2B) and the antibody was thus considered selective for the human-derived PFFs. Staining for α-tubulin and DAPI indicated that after a two-hour exposure, PFFs localised to the cytoplasm and did not enter the nucleus (Figure 2C), with a greater degree of cytoplasmic uptake visible after 24 h (Figure 2D).

### 3.3. R18D Enters STC-1 Cells and Is Non-Toxic

STC-1 cells were incubated with R18D-FITC (5 or 10 μM) for 10 min, fixed and co-stained with DAPI. At both concentrations, R18D-FITC entered STC-1 cells, with diffuse staining throughout the cytoplasm and punctate localisation in the nucleus (Figure 3A). Based on the MTS cell viability assay, a 24-h treatment of STC-1 cells with increasing concentrations of R18D (0.5–10 μM) did not cause toxicity but increased cell metabolic activity in a dose-dependent manner (0.5 μM = 1.01 ± 0.03, 10 μM = 1.12 ± 0.10; fold change relative to untreated controls), however, did not demonstrate statistical significance (Figure 3B).

### 3.4. Pre-Treatment with R18D Reduces PFF Uptake by STC-1 Cells

Treatment of STC-1 cells with R18D before and during a 2 h exposure to PFF-488 prevented internalisation of the protein aggregates, demonstrated by live-cell imaging (Figure 4A) and immunofluorescence (Figure 4B). Additional quantification of immunofluorescence (Figure 4C) determined that R18D significantly prevented PFF-488 uptake (H (2) = 9.629; *p* = 0.001), with multiple comparisons demonstrating that the mean normalised fluorescent intensity of PFF-488 group (2.436 ± 1.137 arbitrary units) was significantly higher than controls (0.004 ± 0.001; *p* = 0.0218) and R18D + PFF-488 (0.004 ± 0.001; *p* = 0.0373). There was no significant difference between R18D + PFF-488 and untreated controls. 

## 4. Discussion

A pro-inflammatory gut environment is implicated in the pathogenesis and progression of PD and peripheral α-synuclein pathology is observed prior to central dopaminergic α-synuclein pathology and motor symptoms in animal models, suggesting an involvement in early disease processes [29]. Moreover, the α-synuclein protein may contribute to gastrointestinal immune responses [8,42] and elevated α-synuclein levels in enteric neurons have been linked to gut inflammation both in people with and without PD [43]. However, despite an interaction between α-synuclein and various luminal factors (e.g., pro-inflammatory microbial metabolites), α-synuclein is not well-characterised in the intestinal epithelial barrier. In this study, we, therefore, investigated the expression and distribution of α-synuclein in an enteroendocrine cell line and subsequently examined a novel therapeutic approach to prevent the internalisation of pathogenic α-synuclein aggregates using the arginine-rich peptide, R18D.

This study supports emerging research that enteroendocrine cells express α-synuclein, as well as pS129-phosphorylated α-synuclein. Embedded in the gut barrier, enteroendocrine cells secrete various peptide hormones in response to chemo- and mechano-sensation, with a well-characterised role in nutrient sensing and control of metabolism and appetite. However, emerging evidence indicates enteroendocrine cells possess neuronal-like features [44] and are also integral for host-microbiota communication, gut homeostasis and inflammation [45,46,47,48]. Due to their physiological characteristics and anatomical position bridging the intestinal barrier and enteric nervous system, it has been hypothesised that enteroendocrine cells are a site of α-synuclein misfolding and aggregation [31]. Supporting previous studies [31,32,33], we demonstrated α-synuclein expression in STC-1 cells (Figure 1). Notably, immunofluorescent microscopy revealed abundant staining of pS129 α-synuclein in the STC-1 cell line. While the function of phosphorylated α-synuclein at serine-129 is still debated [49], studies demonstrate that this post-translational modified protein activates intracellular protein clearance mechanisms in response to cellular stress (e.g., calcium influx or reactive oxygen species) [50] and forms fibrils more readily than de-phosphorylated α-synuclein [51]. Furthermore, it is widely accepted that pS129 α-synuclein is over-abundant in Lewy bodies compared to healthy brains [52] and its immunoreactivity is a highly specific biomarker for PD [49]. As such, the abundance of pS129 α-synuclein in STC-1 cells indicates a potential for pathogenic misfolding and Lewy body formation in a gastrointestinal cell population, especially when coupled with a pro-inflammatory gut environment. 

Under certain conditions, monomeric α-synuclein forms fibrillar aggregates which can be taken up by neighbouring cells, a process modelled through the use of PFFs [53]. We demonstrated that PFFs are internalised by STC-1 cells (Figure 2), mirroring previous research in neurons [54,55,56]. Importantly, our findings support a recent study [57], where PFF uptake by STC-1 cells caused intracellular Ca^2+^ signalling and resulted in the transfer of α-synuclein fibrils to SH-SY5Y cells in co-culture. This process was dependent on cell-to-cell contact, which is intriguing given evidence for synapse-like connections between enteroendocrine cells and neurons [44]. Further studies investigating α-synuclein in non-neuronal cells are scarce; however, administration of PFFs to endothelial cells in vitro reduces the expression of tight junction proteins, especially zona occludens-1 and occludin, leading to functional impairments in permeability [58]. Notably, epithelial α-synuclein can play a causative role in PD, as recent over-expression of human α-synuclein in intestinal stem cells of the *Drosophila* midgut induced various PD-like features, including microbial dysbiosis, disrupted intestinal homeostasis, dopaminergic neuronal death and motor defects [59]. Importantly, these intestinal stem cells differentiate into both absorptive enterocytes and secretory enteroendocrine cells [59]. These studies, coupled with the leaky and inflamed gut evident in people with PD, highlight the need to investigate the complex interactions between α-synuclein aggregation, enteroendocrine cells and gut inflammation, which could then induce α-synuclein pathology in enteric neurons.

Having confirmed that enteroendocrine cells express endogenous α-synuclein and can internalise PFFs, coupled with the known exposure of enteroendocrine cells to a pro-inflammatory gut environment which may promote α-synuclein aggregation, this study investigated the potential of the poly-arginine peptide R18D to prevent the intercellular spread of α-synuclein pathology. Firstly, the study demonstrated that R18D is readily internalised by STC-1 cells, with diffuse cytoplasmic and punctate nuclear localisation (Figure 3). This is not surprising given the known cell-penetrating properties of cationic-arginine-rich peptides and previous studies demonstrating neuronal cell uptake of R18D L-enantiomer (R18; [60]. Importantly, we showed that R18D had a marked inhibitory effect on PFF internalisation in STC-1 cells (Figure 4); however, the mechanisms of PFF uptake and R18D action are both areas of ongoing research. 

Various mechanisms may underlie the entry of PFFs into cells, with evidence for clathrin-mediated [61] and caveolae-dependent endocytosis [62], as well as direct membrane permeation [63,64]. In a recent study using an epithelial cell line (U-2 OS), iPSC-derived human dopaminergic neurons and astrocytes demonstrated that PFFs are rapidly internalised and transferred to lysosomes through micropinocytosis, a separate process to clathrin-mediated endocytosis that is not receptor-based and does not rely on early or recycling endosomes [65]. Similarly, a study using induced pluripotent stem cell-derived neurons identified an internalisation pathway involving the transmembrane protein glycoprotein nonmetastatic melanoma protein B (GPNMB) [66]. Taken together, it is likely that different cells possess the capacity to internalise PFFs using different cellular uptake pathways, as occurs with cell-penetrating peptides [67].

With respect to the mechanism of how R18D inhibits α-synuclein PFFs internalisation, we propose two main possibilities. Given that PFFs are negatively charged and R18D is positively charged, the two molecules may bind through electrostatic interactions, resulting in a charge-neutral compound that inhibits the anionic-dependent uptake mechanisms of the PFFs. Similarly, recent work demonstrated that a positively charged cationic peptide (LL-37, a 37-mer endogenous anti-microbial peptide with 5 arginine residues) inhibited the formation of α-synuclein amyloids [68]. LL-37 has a net charge of +6 and thus, it is likely R18D would confer an even greater benefit in inhibiting the formation of α-synuclein amyloids due to its higher arginine content and net charge (+18 charge). 

Secondly, R18D could compete for cellular uptake pathways utilised by the PFFs, especially given that the STC-1 cells were pre-incubated with the peptide prior to the addition of PFFs to the cells. Notably, emerging research demonstrates the importance of heparin sulphate proteoglycans (HSPGs) in the propagation and uptake of proteins involved in neurodegeneration, such as α-synuclein [69,70]. Specifically, the internalisation of α-synuclein fibrils was dependent on HSPG in both neuroblastoma and oligodendrocyte cell lines in vitro [71], while PFF-induced toxicity, uptake and motor dysfunction depend on the HSPG pathway in *Caenorhabditis elegans* [69]. Importantly, CARPs like R18D form electrostatic interactions with anionic chemical moieties located within membrane proteoglycans, such as HSPGs [34] and may thus interfere with HSPG-mediated PFF uptake. However, once added to the cells, the PFFs would potentially neutralise the cationic charge of R18D and thereby inhibit the capacity of the peptide to engage internalisation pathways. Under certain physiological conditions, monomeric α-synuclein forms fibrillar aggregates, which interact with other monomeric α-synuclein proteins through transient electrostatic interactions to cause further aggregation [72,73]. Recent research from our laboratory demonstrates the poly-arginine-18 peptide (R18; L-enantiomer of R18D) can inhibit lysosomal protein misfolding [74] possibly by interrupting electrostatic interactions between proteins [34]. As such, regardless of the mechanism preventing PFF uptake, R18D may also have the capacity to inhibit α-synuclein fibril formation and thereby indirectly reduce the spread of protein aggregates.

Finally, it is worth mentioning that given gastrointestinal immune responses may contribute to α-synuclein aggregation and fibril formation [75,76], R18D may also have the capacity to mitigate inflammation in the gut. If so, this provides an additional mechanism whereby R18D could help reduce α-synuclein aggregation and spread in vivo. The anti-inflammatory properties of CARPs are well-established [34]. For example, several CARPs have been shown to inhibit the activation of NF-kB, which could reduce the expression of genes involved in pro-inflammatory pathways. Other CARPs have been shown to inhibit the expression of pro-inflammatory cytokines (e.g., TNF-α, IL-6) and reduce cell surface receptors involved in inflammation (e.g., TLR4, TNFR).

Further studies are necessary to investigate how R18D affects both fibrillar aggregates and endogenous monomeric α-synuclein in vitro and in vivo, particularly given the widespread expression of the protein throughout the body and its’ various roles [77,78]. Moreover, it is important to acknowledge that targeting PD pathology is very complex, particularly when considering the progressive nature of the disease, the contribution of other proteins to Lewy body pathology as well as the disruption of various cellular and molecular processes throughout the disease course. However, the results discussed herein demonstrate the potential of R18D to inhibit cellular uptake of PFFs and warrant further investigation. 

## 5. Conclusions

This study has contributed to the growing evidence concerning the presence of α-synuclein in gastrointestinal cells and has demonstrated the therapeutic potential of R18D to inhibit the internalisation of PFFs. Future studies are required to investigate how an altered gut environment may influence α-synuclein conformation and aggregation in enteroendocrine cells and if this contributes to pathogenic α-synuclein pathology in enteric neurons and along the gut–brain axis. In addition, as cationic arginine-rich peptides have the capacity to reduce protein misfolding, it is imperative to investigate if R18D can also inhibit intracellular α-synuclein protein aggregation. Considering the present findings, R18D represents an exciting potential therapeutic avenue to inhibit the aggregation and spread of α-synuclein and to reduce peripheral neuroinflammation via oral drug delivery to the gut, an important target organ for PD. 

## Figures and Tables

**Figure 1 biomedicines-11-02089-f001:**
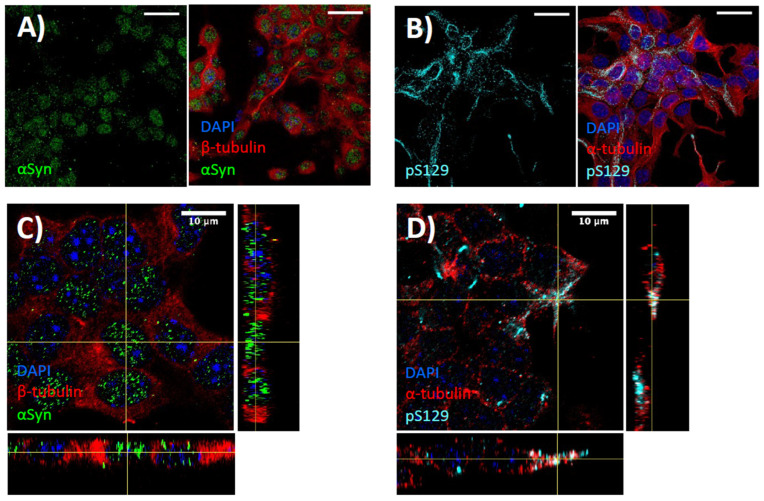
**Expression of α-synuclein in enteroendocrine STC-1 cells in vitro.** Z-stacks were captured at 100× magnification, with maximum intensity projection (**A**,**B**) and a zoomed orthogonal view (**C**,**D**) displayed. In STC-1 cells, α-synuclein (αSyn) expression is limited to the nucleus (co-localising with DAPI staining). Phosphorylated α-synuclein (pS129) forms distinct fibrillar aggregates along the cellular processes of STC-1 cells, identified by co-localisation with α-tubulin staining. Images are representative photos of three independent experiments; the scale bar represents 25 μm (**A**,**B**) and 10 μm (**C**,**D**). For all sets of orthogonal view images (**C**,**D**), the large image shows the x–y view, the bottom image shows the z–x and the right image shows the z–y.

**Figure 2 biomedicines-11-02089-f002:**
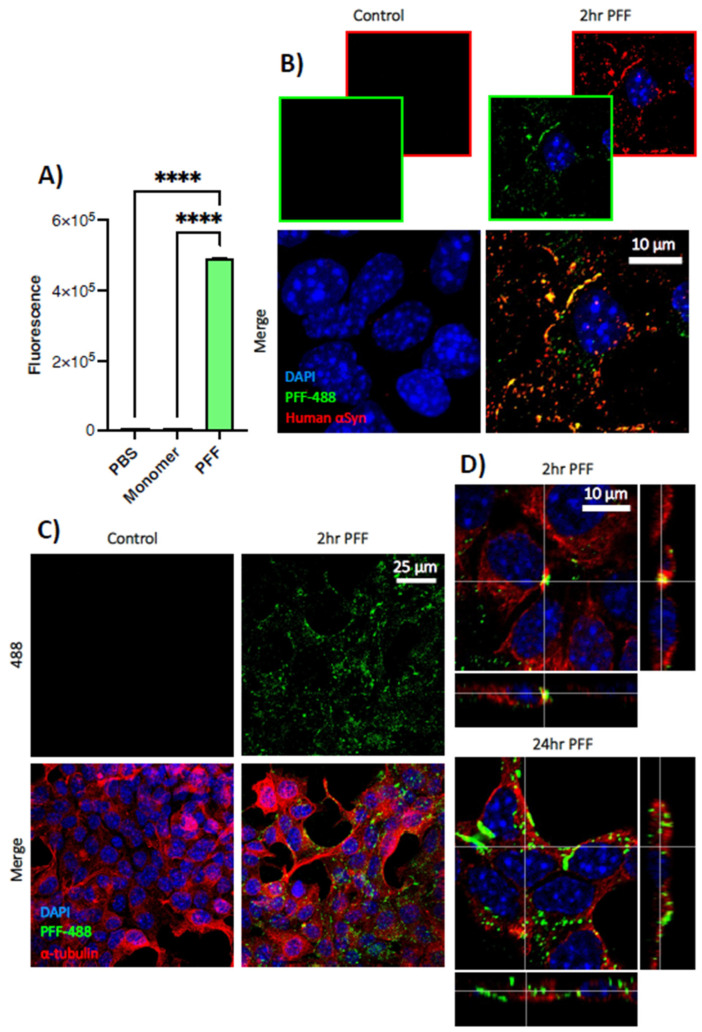
**Pre-formed fibrils of α-synuclein enter enteroendocrine STC-1 cells in vitro.** STC-1 cells were incubated with PFFs tagged with green fluorescent protein (PFF-488) for two hours and then fixed for immunofluorescent staining. (**A**) Formation of α-synuclein pre-formed fibrils (PFFs) was confirmed through thioflavin-T assay, which demonstrated significantly higher mean fluorescence compared to both a PBS control and monomeric α-synuclein protein. **** *p* < 0.0001 (**B**) Most PFF-488 co-localises with human α-synuclein immunoreactivity, indicating successful labelling of PFFs with Alexa Fluor 488. Control STC-1 cells do not demonstrate any positive human α-synuclein staining. (**C**) Co-staining with nuclear marker DAPI and cytoplasmic α-tubulin demonstrates that PFF-488s do not enter the nucleus, rather are concentrated in the cytoplasm. (**D**) Orthogonal views show PFF-488 enters the cell after two hours, with a greater degree of cytoplasmic uptake after 24 h. Figure demonstrates maximum intensity projection of z-stacks imaged at 100× magnification (**B**,**C**) and a magnified orthogonal view (**D**), with scale bars representing 10 μm (**B**,**D**) or 25 μm (**C**). For both sets of orthogonal view images (**D**), the large image shows the x–y view, the bottom image shows the z–x and the right image shows the z–y.

**Figure 3 biomedicines-11-02089-f003:**
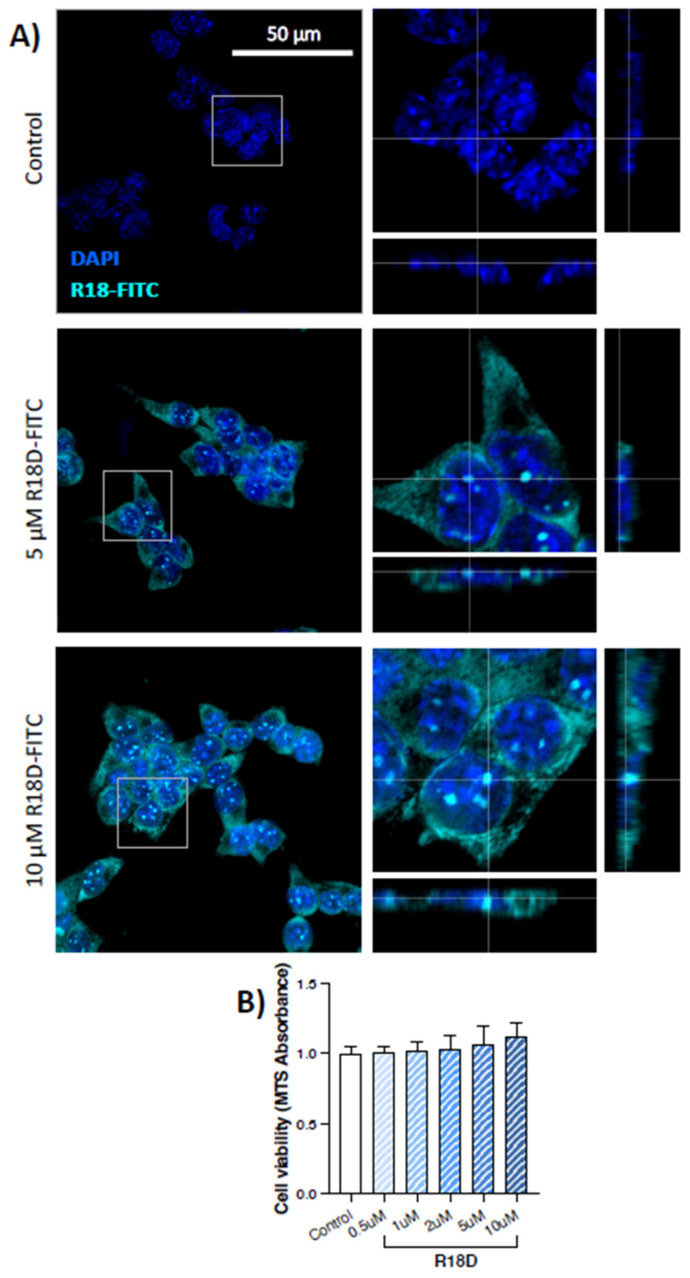
**Cationic arginine-rich peptide R18D is taken up by STC-1 cells in vitro.** (**A**) R18D tagged with a fluorescein isothiocyanate tag (R18D-FITC) enters STC-1 cells after 10 min at both 5 μM and 10 μM. Images are maximum intensity projections of confocal z-stacks acquired at 100× magnification (scale bar represents 50 μm), with the magnified orthogonal view of the boxed area shown beside. (**B**) MTS assay of increasing R18D concentrations (0.5 μM–10 μM) demonstrated a non-significant increase in cell metabolic activity compared to untreated STC-1 cells. For all three sets of orthogonal view images (**A**), the large image shows the x–y view, the bottom image shows the z–x and the right image shows the z–y.

**Figure 4 biomedicines-11-02089-f004:**
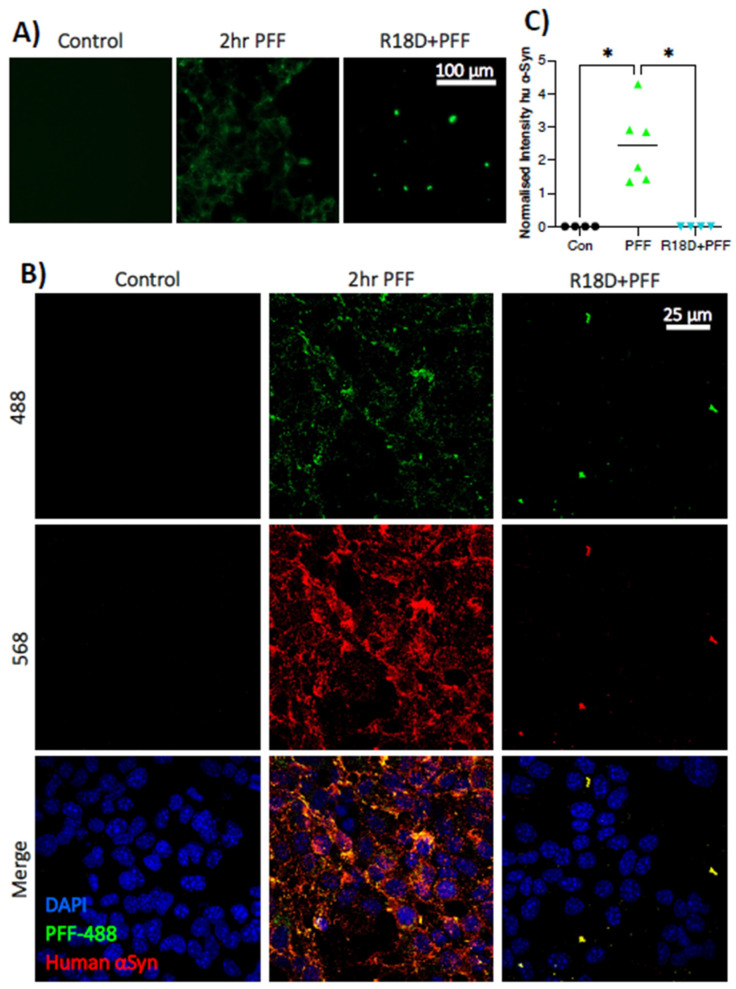
**Cationic arginine-rich peptide R18D prevents PFF-488 uptake by STC-1 cells in vitro.** STC-1 cells were pre-treated with R18D for 10 min prior to and during incubation with PFF-488 for two hours. (**A**) Live cell imaging (20× magnification) demonstrates that R18D prevents PFF uptake (scale bar represents 100 μm). (**B**) Media was aspirated after two hours and cells were fixed and immunostained for human α-synuclein and DAPI for confocal microscopy. Treatment with R18D clearly prevents cellular uptake compared to the PFF-488 group. Images are maximum intensity projections of confocal z-stacks acquired at 100× magnification (scale bar represents 25 μm). (**C**) The mean fluorescence intensity of human α-synuclein was significantly higher in the PFF-488 group compared to untreated controls and the PFF-R18D group, * *p* < 0.05.

## Data Availability

The data presented in this study are available on request from the corresponding author.

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
