# Peer review of "The Poly-Arginine Peptide R18D Interferes with the Internalisation of α-Synuclein Pre-Formed Fibrils in STC-1 Enteroendocrine Cells"

_biomedicines, 2023, doi:10.3390/biomedicines11082089_

Round 1
Reviewer 1 Report
Review of the manuscript entitled: The poly-arginine peptide R18D interferes with the internalization of α-Synuclein pre-formed fibrils in STC-1 enteroendocrine cells.
1. In abstract and introduction clear aim of the manuscript should be added e.g. "The aim of the present study was to ...". In case of introduction aim should be at the end of introduction.
2. The manuscript is very messy, please standardize the fonts because they are different sizes and styles. E.g line 78 and 85 please correct the rest of the manuscript.
3. The introduction, methodology, results and discussion are well written. However, I would suggest adding more details and describing the results in more detail. please add in the description of the results by what percentage the values changed.
4. Use higher quality photos if possible.
5. In the discussion, it may be appropriate to add information, how other types of peptides affect the development or lack of inflammation?
Reviewer 2 Report
The authors studied the endogenous expression of alph-synuclein in mouse enteroendocrine (STC-1) cells and found that STC-1 cells pretreated with the cationic arginine-rich peptide R18D interferred with uptake of pre-formed α-synuclein fibrils (PFFs) in vitro. Please see the attached pdf file for comments.
